# Crowding-induced phase separation of nuclear transport receptors in FG nucleoporin assemblies

Luke K Davis[1,2,3]*[†], Ian J Ford[1,2,3], Bart W Hoogenboom[1,2,3]*

[1]Department of Physics and Astronomy, University College London, London, United Kingdom; [2]Institute for the Physics of Living Systems, University College London, London, United Kingdom; [3]London Centre for Nanotechnology, University College London, London, United Kingdom

**\*For correspondence:**
luke.davis@uni.lu (LKD);
b.hoogenboom@ucl.ac.uk (BWH)

**Present address:** [†]Department of Physics and Materials, University of Luxembourg, Limpertsberg, Luxembourg

**Competing interest:** The authors declare that no competing interests exist.

**Abstract** The rapid (<1 ms) transport of biological material to and from the cell nucleus is regulated by the nuclear pore complex (NPC). At the core of the NPC is a permeability barrier consisting of intrinsically disordered phenylalanine-glycine nucleoporins (FG Nups). Various types of nuclear transport receptors (NTRs) facilitate transport by partitioning in the FG Nup assembly, overcoming the barrier by their affinity to the FG Nups, and comprise a significant fraction of proteins in the NPC barrier. In previous work (Zahn et al., 2016), we revealed a universal physical behaviour in the experimentally observed binding of two well-characterised NTRs, Nuclear Transport Factor 2 (NTF2) and the larger Importin-β (Imp-β), to different planar assemblies of FG Nups, with the binding behaviour defined by negative cooperativity. This was further validated by a minimal physical model that treated the FG Nups as flexible homopolymers and the NTRs as uniformly cohesive spheres. Here, we build upon our original study by first parametrising our model to experimental data, and next predicting the effects of crowding by different types of NTRs. We show how varying the amounts of one type of NTR modulates how the other NTR penetrates the FG Nup assembly. Notably, at similar and physiologically relevant NTR concentrations, our model predicts demixed phases of NTF2 and Imp-β within the FG Nup assembly. The functional implication of NTR phase separation is that NPCs may sustain separate transport pathways that are determined by inter-NTR competition.

## Editor's evaluation

This theoretical study describes the interaction of a planar brush or film of the resident unstructured components of the nuclear pore complex (NPC) called nucleoporins (FG-nups) and different nuclear transport receptors (NTRs). The authors describe impacts of competitive binding that give rise to enrichment of the NTRs, NTF2 and importin-β, at different depths of the FG-nup film, which could relate to experimental observations in other studies, as well as evidence that crowding could promote the rate of nuclear transport by modulating FG-NTR binding/unbinding. The conclusions were found to be generally supported by the data, relevant to the field of nuclear transport, and able to make specific predictions that can be experimentally tested in the future.

## Introduction

Embedded in the nuclear envelope are nuclear pore complexes (NPCs), large hour-glass shaped channels (inner diameter ~40 nm) that regulate biomolecular transport between the cytoplasm and nucleoplasm (*Alberts, 1994*; *Wente, 2000*). The NPC presents an exclusion barrier to inert molecules, with the degree of exclusion increasing with molecular size. This physical barrier arises from a dense (mass

density 100–300 mg/ml) assembly of moderately cohesive intrinsically disordered phenylalanine-glycine nucleoproteins (FG Nups; *Hoogenboom et al., 2021*). In addition, the barrier contains relatively high numbers (~20–100) of nuclear transport receptors (NTRs), globular proteins that facilitate the translocation of cargo by transiently binding to the FG Nups (*Lowe et al., 2015*; *Kim et al., 2018*; *Hayama et al., 2018*). The known roles of NTRs in nucleocytoplasmic transport include ferrying specific cargo in and/or out of the nucleoplasm, returning RanGTP to the cytoplasm, and enhancing the exclusion of inert molecules (*Jovanovic-Talisman et al., 2009*; *Aitchison and Rout, 2012*; *Lowe et al., 2015*; *Jovanovic-Talisman and Zilman, 2017*; *Kapinos et al., 2017*). However, it remains to be fully elucidated how different NTRs organise themselves within the permeability barrier and how this organisation affects transport (*Stanley et al., 2017*; *Jovanovic-Talisman and Zilman, 2017*; *Hoogenboom et al., 2021*).

Qualitatively, NTRs are required to facilitate cargo transport, yet their presence in high numbers poses a significant risk of jamming the transport channel due to crowding effects (*Hoogenboom et al., 2021*). These seemingly contradictory phenomena have inspired various propositions about more subtle roles of NTRs in the NPC, such as their being essential to maintaining the barrier to non-specific transport; and the existence of spatially segregated, separate transport pathways through the NPC, where different NTRs and/or cargoes may take different trajectories through the barrier (*Shah and Forbes, 1998*; *Yang and Musser, 2006*; *Naim et al., 2007*; *Fiserova et al., 2010*; *Yamada et al., 2010*; *Ma et al., 2012*; *Kapinos et al., 2014*; *Lowe et al., 2015*; *Lim et al., 2015*; *Ma et al., 2016*; *Kapinos et al., 2017*; *Perez Sirkin et al., 2021*). Furthermore, NTRs of varying size and affinity to the FG Nups may play different roles in maintaining efficient transport, e.g., some smaller and more cohesive NTRs may play the role of a cross-linker, modulating the distribution of FG Nup mass in the pore, thereby influencing the trajectories of larger NTRs or cargoes (*Perez Sirkin et al., 2021*). An alternative mechanism, involving the switching between import and export transport states, has also been proposed (*Kapon et al., 2008*). Another observation to keep in mind is that the apparent binding affinities of NTRs to FG Nups in vitro appear too tight to enable the fast transport as seen in native NPCs; this can be strongly modulated, however, by the presence of other cellular proteins that compete – non-specifically – with the NTRs close to the FG Nup mass (*Tetenbaum-Novatt et al., 2012*; *Lennon et al., 2021*).

It is difficult to test different hypotheses regarding how NTR crowding affects the NPC barrier in an in vivo setting, due to the complexity of probing multitudes of diverse proteins in a dense nanoscale channel, as is the case in the NPC. To circumvent this complexity, various studies have reverted to much simpler in vitro FG Nup and NTR assemblies that resemble the physical environment of the NPC, e.g., considering the behaviour of FG Nups in solutions or assemblies with a similar mass density of FG Nups as found in the NPC (~200 mg/ml) (*Ghavami et al., 2016*; *Davis et al., 2020*). Particularly well-controlled model systems are polymer film assemblies, where copies of an FG Nup are anchored to a planar surface and where NTRs (typically of one type) are introduced in the bulk volume above the surface (*Eisele et al., 2010*; *Schoch et al., 2012*; *Eisele et al., 2012*; *Kapinos et al., 2014*; *Zahn et al., 2016*; *Vovk et al., 2016*). It merits emphasising that FG Nup polymer film assemblies are but a simplified model for the NPC in that, typically, only one type of FG Nup and one type of NTR are probed, whereas in the NPC there are multiple different types of FG Nups and NTRs in the inner channel; and in that – in polymer film assemblies – the geometry of the nanoconfinement of the FG Nups and NTRs arises from a single hard planar wall, as compared with the more complex cylindrical confinement in the NPC. Nonetheless, they provide a most suitable platform to discover general principles of FG Nup and NTR behaviour that may be obscured in experiments on in vivo NPCs; and provide a foundation for a step-by-step increase in the level of complexity towards that of real NPCs. For the behaviour of planar films containing one type of FG Nup and one type of NTR, the main findings thus far have been: (1) that NTRs of one type (such as NTF2 and Importin-β [Imp-β]) bind to FG Nups in a rather generic way, suggesting universal physical principles – based on negative cooperative binding – might govern their behaviour (*Vovk et al., 2016*; *Zahn et al., 2016*); (2) NTRs readily penetrate the FG Nup films, with only limited effects on the collective morphology of the FG Nups (little swelling or compaction) (*Eisele et al., 2010*; *Kapinos et al., 2014*; *Wagner et al., 2015*; *Vovk et al., 2016*; *Zahn et al., 2016*); (3) that such systems can replicate the basic selective mechanism in the NPC, i.e., inert proteins tend not to penetrate the collective FG Nup phases but NTRs do, consistent with in vivo NPC functionality and with experiments on bulk solutions of FG Nups and NTRs (*Schmidt*

*and Görlich, 2015*; *Schmidt and Görlich, 2016*); (4) the number of transport proteins in the FG Nup films can vary by orders of magnitude as a function of NTR numbers in solution above the film, in a highly non-Langmuir manner, where complex many-body interactions preclude the use of simple one-to-one binding models (*Eisele et al., 2010*; *Schoch et al., 2012*; *Kapinos et al., 2014*; *Wagner et al., 2015*; *Vovk et al., 2016*; *Zahn et al., 2016*). With the caveat that only a subset of NTRs have been probed, investigations of planar assemblies of FG Nups and NTRs highlight the fine-tuned balance of FG Nup attachment density, FG Nup cohesion, FG Nup-NTR affinities, the amount of non-specific proteins in the system, and NTR concentrations, where minor changes in this balance can lead to qualitatively different binding scenarios (*Vovk et al., 2016*; *Zahn et al., 2016*; *Stanley et al., 2017*; *Lennon et al., 2021*).

Taking a next step towards the complexity of the in vivo NPC and considering the large number (~20–100) of NTRs of different sizes and affinities in the NPC inner channel (*Lowe et al., 2015*; *Peters, 2009a*; *Peters, 2009b*; *Lim et al., 2015*; *Kim et al., 2018*; *Hoogenboom et al., 2021*), one may next ask how the binding affinity of a specific NTR to a planar assembly of FG Nups depends on the binding behaviour of other NTRs. Physiologically, the answers to this question may explain how binding and thereby transport of specific NTRs, and their associated cargoes, can be modulated (if at all) by the presence of other types of NTRs, since this would directly impact on the transport function of the NPC. More generally, such answers will aid our understanding of how the NPC maintains fast and efficient transport whilst hosting a dense milieu of FG Nup motifs, NTRs, and cargoes. Finally, a systematically probing of crowding effects provides a means to assess various conceptual models of NPC transport such as the 'Kap-centric' barrier model (*Wagner et al., 2015*) and reduction-of-dimensionality models (*Peters, 2005*).

Here, we aim to understand how the crowding by different NTRs may affect the organisation and thereby transport functionality of FG Nup assemblies. Focusing on planar FG Nup assemblies as a well-controlled model system (*Zahn et al., 2016*), we use physical modelling to probe how the binding of one type of NTR could be affected by other types of NTRs, in a way that can be tested by currently available experimental setups. To explore the effects of mixed NTR crowding, we model a ternary mixture containing two different NTRs and one type of FG Nup in a polymer film assembly. When modelling FG Nups and NTRs, one can take various coarse-grained approaches, for instance one can take an all-atom approach (*Miao and Schulten, 2009*; *Gamini et al., 2014*; *Raveh et al., 2016*), or account only for the amino acids (*Ghavami et al., 2013*, *Ghavami et al., 2014*; *Ghavami et al., 2018*), or work only with the generic patterning of FG/hydrophobic/hydrophilic/charged 'patches' (*Tagli-azucchi and Szleifer, 2015*; *Matsuda and Mofrad, 2021*; *Davis et al., 2021*), or completely neglect sequence details altogether in a 'homopolymer' approach (*Moussavi-Baygi et al., 2011*; *Osmanovic et al., 2012*; *Osmanović et al., 2013b*; *Vovk et al., 2016*; *Zahn et al., 2016*; *Timney et al., 2016*; *Davis et al., 2020*). Each approach has its strengths and weaknesses. For instance higher-resolution modelling can account for greater molecular complexity, but with the difficulty in probing large time and length scales. In contrast, homopolymer modelling provides access to larger time and length scales with more robust parameterisation and simplicity of execution, at the expense of (sub)molecular detail. In this work, we build on our previous minimal modelling framework based on treating FG Nups as sticky and flexible homopolymers and NTRs as uniformly cohesive spheres (*Osmanović et al., 2013b*; *Zahn et al., 2016*; *Davis et al., 2020*).

## Results

Minimal physical modelling facilitates the understanding of many aspects of NPC functionality in terms of general principles, but it requires quantitatively accurate parameter settings to make meaningful predictions (*Osmanović et al., 2013a*; *Jovanovic-Talisman and Zilman, 2017*; *Hoogenboom et al., 2021*). In this work, the minimal modelling framework we employ is that of coarse-grained classical density functional theory (DFT), that has been previously used to model aspects of the NPC permeability barrier (*Osmanović et al., 2013b*; *Zahn et al., 2016*; *Davis et al., 2020*). To ensure that the setting of the parameters in our DFT model – outlined in the Computational methods section (below) – is commensurate with the behaviour of FG Nups (Nsp1) and NTRs (NTF2 or Imp-β) as probed in experiments, we make use of experimental data on FG Nup-NTR polymer film assemblies where the macroscopic binding between one type of FG Nup (Nsp1) and one type of NTR (NTF2 or Imp-β) was quantitatively probed (see *Figure 1*) *Zahn et al., 2016*. The experiments focused on a polymer

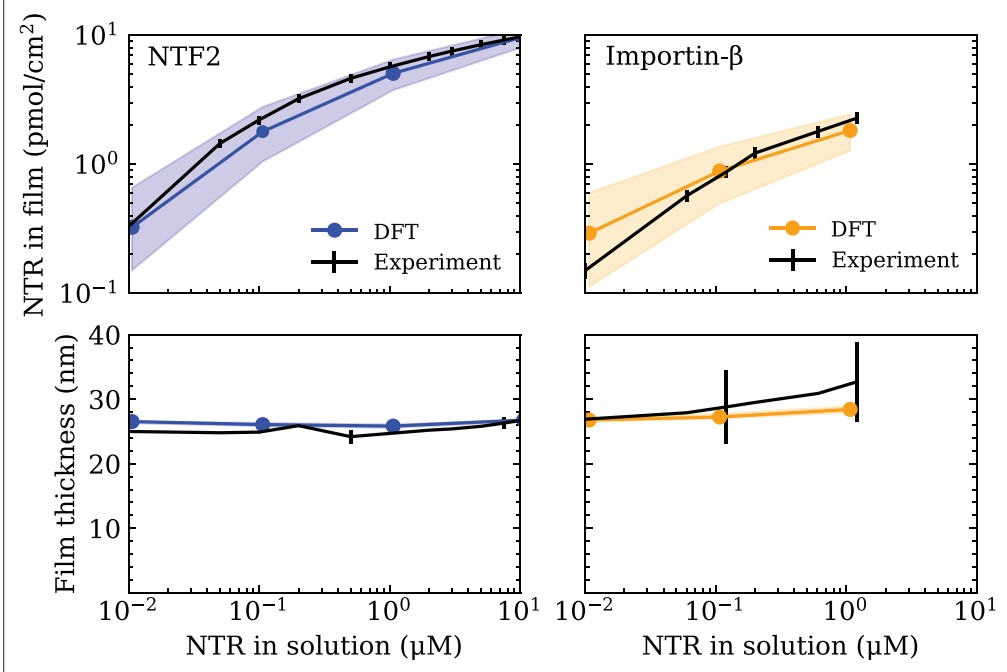

**Figure 1.** Setting the polymer-particle cohesion strengths $\{\epsilon_{\text{FG-NTF2}}, \epsilon_{\text{FG-Imp}\beta}\}$ through comparison of density functional theory (DFT) results with experimental binding isotherms for the cases of NTF2 (left) and Importin-$\beta$ (Imp-β) (right) binding to an Nsp1 film (*Zahn et al., 2016*) (top). Concomitant film thicknesses as found in DFT and experiment (bottom). The experimental Nsp1 surface attachment densities were 4.9 pmol/cm² and 5.1 pmol/cm² for NTF2 and Imp-β, respectively. The parametrised cohesion strengths $\epsilon_{\text{FG-NTF2}} = 2.4 \pm 0.1\ k_{\text{B}}T$ and $\epsilon_{\text{FG-Imp}\beta} = 2.3 \pm 0.1\ k_{\text{B}}T$ correspond to the modelled NTF2 and Imp-β particles, respectively. Filled bands (in all four panels) denote a tolerance of ±0.1 $k_{\text{B}}T$ in the polymer-particle cohesion strengths. The thicknesses of the filled bands for the bottom two panels are similar to the thickness of the line connecting DFT data points (blue and orange).

The online version of this article includes the following source code and figure supplement(s) for figure 1:

**Source code 1.** Simulation parameters for the classical density functional theory code.

**Figure supplement 1.** Parameterising the polymer cohesion strength $\epsilon_{\text{FG-FG}}$.

**Figure supplement 2.** Inert particles of growing size do not penetrate the polymer film.

film consisting of Nsp1 domains, attached to a flat surface at physiologically relevant densities (≈3.3 polymers/100 nm²), interacting with either NTF2 or Imp-β over a physiologically relevant range of concentrations (0.01–10 µM).

For consistency with the available experimental data we focus on the FG Nup Nsp1, which we treat as a homogeneous, flexible, and cohesive polymer consisting of $M = 300$ beads of diameter $d^{(3)} = 0.76$ nm (two amino acids per bead), resulting in the approximately correct persistence length for FG Nups (*Lim et al., 2006*; *Zahn et al., 2016*; *Hayama et al., 2019*; *Davis et al., 2020*). The intermolecular and intramolecular cohesive properties of FG Nups arise from a combination of hydrophobic motifs, e.g., FG, FxFG, and GLFG, and charged residues along the sequence which, in our model, are subsumed into one single cohesion parameter $\epsilon_{\text{FG-FG}}$ (equivalent to $\epsilon^{(33)}$ in the Computational methods section). In addition to the FG Nups, we also include the presence of the NTRs, NTF2 and Imp-$\beta$, which we model as uniformly cohesive spheres of diameters $d_{\text{NTF2}} = 4$ nm (same as $d^{(1)}$ in the Computational methods section) and $d_{\text{Imp-}\beta} = 6$ nm (same as $d^{(2)}$ in the Computational methods section), respectively (*Zahn et al., 2016*). The cohesive properties of the NTRs strictly refer to the attraction between the NTRs and FG Nups, arising at least in part from the hydrophobic grooves and charged regions on the former and the hydrophobic motifs and charged regions on the latter (*Kumeta et al., 2012*; *Kim et al., 2013*; *Hayama et al., 2018*; *Frey et al., 2018*). We do not consider any attractive interactions between NTRs themselves as there is no empirical evidence that suggests there are cohesive interactions between NTRs, a choice that is consistent with previous work and shown to well replicate

experimental data of NTR binding to FG Nup assemblies (*Osmanović et al., 2013b*; *Zahn et al., 2016*; *Vovk et al., 2016*). As with the FG Nup intermolecular and intramolecular cohesive interactions, we subsume all contributions to the respective cohesive interactions FG Nup – NTF2 and FG Nup – Imp-$\beta$ through two more cohesion parameters $\epsilon_{\text{FG-NTF2}}$ and $\epsilon_{\text{FG-Imp}\beta}$ (equivalent to the cohesion variables $\epsilon^{(12)}$ and $\epsilon^{(13)}$, respectively).

We begin the parametrisation of our model with the setting of $\epsilon_{\text{FG-FG}}$ so as to accurately reproduce the experimental thickness of Nsp1 planar film assemblies, at similar anchoring densities, as was done previously (*Zahn et al., 2016*; *Fisher et al., 2018*; *Davis et al., 2020*) (see *Figure 1—figure supplement 1*). With the here chosen interaction potential, the resulting cohesion strength is $\epsilon_{\text{FG-FG}} = 0.275 \pm 0.025\, k_{\text{B}}T$ (with experiments conducted at $\approx 23\,^{\circ}\text{C}$), that yields a film thickness $\tau = 26 \pm 2$ nm, in our model defined as the height above the surface below which 95% of the total polymer density is included. We note that the value of $\epsilon_{\text{FG-FG}}$ found here is different to that of our previous work (*Zahn et al., 2016*), mainly due to the different choice of interaction potential. Nonetheless, both the model here and the model in *Zahn et al., 2016* are parametrised using the same experimental data and produce the same film thicknesses. To further validate this value of $\epsilon_{\text{FG-FG}}$, we checked whether the polymer film would exclude inert molecules, a basic property that has been observed for Nsp1 assemblies (*Schmidt and Görlich, 2015*; *Schmidt and Görlich, 2016*). The inert molecules are modelled as non-cohesive spheres of diameter $d^{(i)}$, with $i$ labelling the particle type, and their inclusion/exclusion in the film is quantified through the potential of mean force (PMF) $W^{(i)}(z)$ given as

$$
\begin{aligned}
W^{(i)}(z) &= c^{(i)}(z) + V_{ext}^{(i)}(z) + \int \rho^{(3)}(z)u^{(i)}(z-z')\mathrm{d}z' - \mu^{(i)}, \\
&\approx -k_{\text{B}}T \ln\left(\frac{\rho^{(i)}(z)}{\rho_{\text{bulk}}^{(i)}}\right),
\end{aligned}
\tag{1}
$$

where $c^{(i)}(z)$ is the one-body direct correlation function (see *equation 19*), $V_{ext}^{(i)}(z)$ is the external potential (see *equation 11*), $\rho^{(3)}(z)$ is the polymer number density, $u^{(i)}(z)$ is the one-dimensional (1D), i.e., integrated over the $x-y$ plane, polymer-particle cohesive pair potential (see *equation 13*), and $\mu_{\text{exc}}^{(i)}$ is the excess chemical potential (here set to 0 $k_{\text{B}}T$) (*Roth et al., 2000*; *Roth and Kinoshita, 2006*). For the inert molecules, the polymer-particle attraction (third term in *equation (1)*) is nullified. As expected, non-cohesive particles with increasing diameters (1.0, 2.0, 4.0, and 6.0 nm) experience a greater potential barrier upon attempted entry into the polymer film (see *Figure 1—figure supplement 2*), confirming that our simple model of an Nsp1 film replicates one of the key characteristics of the permeability barrier as seen in the NPC: the degree of exclusion of inert molecules increasing with molecular size (*Mohr et al., 2009*; *Popken et al., 2015*; *Ghavami et al., 2016*). We note that the presence of a maximum, close to the anchoring surface, in the relative density for inert particle diameters $d = 1$ and 2 nm is due to the decrease in the polymer density closer to the surface (consistent with a small potential well close to the surface, see *Figure 1—figure supplement 2b*); the appearance of the maxima in the density depends upon the anchoring density of the FG Nups (not shown here).

We find that particles of diameters $\geq 4$ nm experience PMFs of order 10 $k_{\text{B}}T$. In experiments, the size exclusion limit of the NPC was determined as ~5 nm, albeit that this limit is rather soft and gradual (*Keminer and Peters, 1999*; *Paine et al., 1975*; *Mohr et al., 2009*). In our model, this empirical size limit corresponds to PMFs at least one order of magnitude greater than the thermal background 1 $k_{\text{B}}T$ (*Figure 1—figure supplement 2b*). This order of magnitude of energetic barrier is therefore indicative of a physiologically relevant barrier, and is consistent with another – independent – modelling work that explicitly takes into account the amino acid sequence of the FG Nups in a pore geometry (*Ghavami et al., 2016*), as well as with a more recent numerical study that investigated the free energy barriers of nanoparticles of varying size and avidity (*Matsuda and Mofrad, 2021*).

Having shown that the now parametrised polymer model for Nsp1 films replicates the experimentally observed film thickness and the size selectivity of the NPC, we shift our focus to setting the parameters for the NTRs, NTF2 and Imp-$\beta$. The cohesion strengths $\epsilon_{\text{FG-NTF2}}$ (same as $\epsilon^{(13)}$ in the Computational methods section) and $\epsilon_{\text{FG-Imp}\beta}$ (same as $\epsilon^{(23)}$ in the Computational methods section) are set by comparing absorption isotherms as calculated in DFT to those measured in experiment (*Zahn et al., 2016*) (see *Figure 1*). Using DFT, we compute the total density of NTRs in the film $\Gamma^{(i)}$, $i = \{1, 2\}$, as the total NTR population within the film thickness $\tau$ divided by the area $A = 88.62 \times 88.62$ nm$^2$ (converted to units of pmol/cm$^2$).

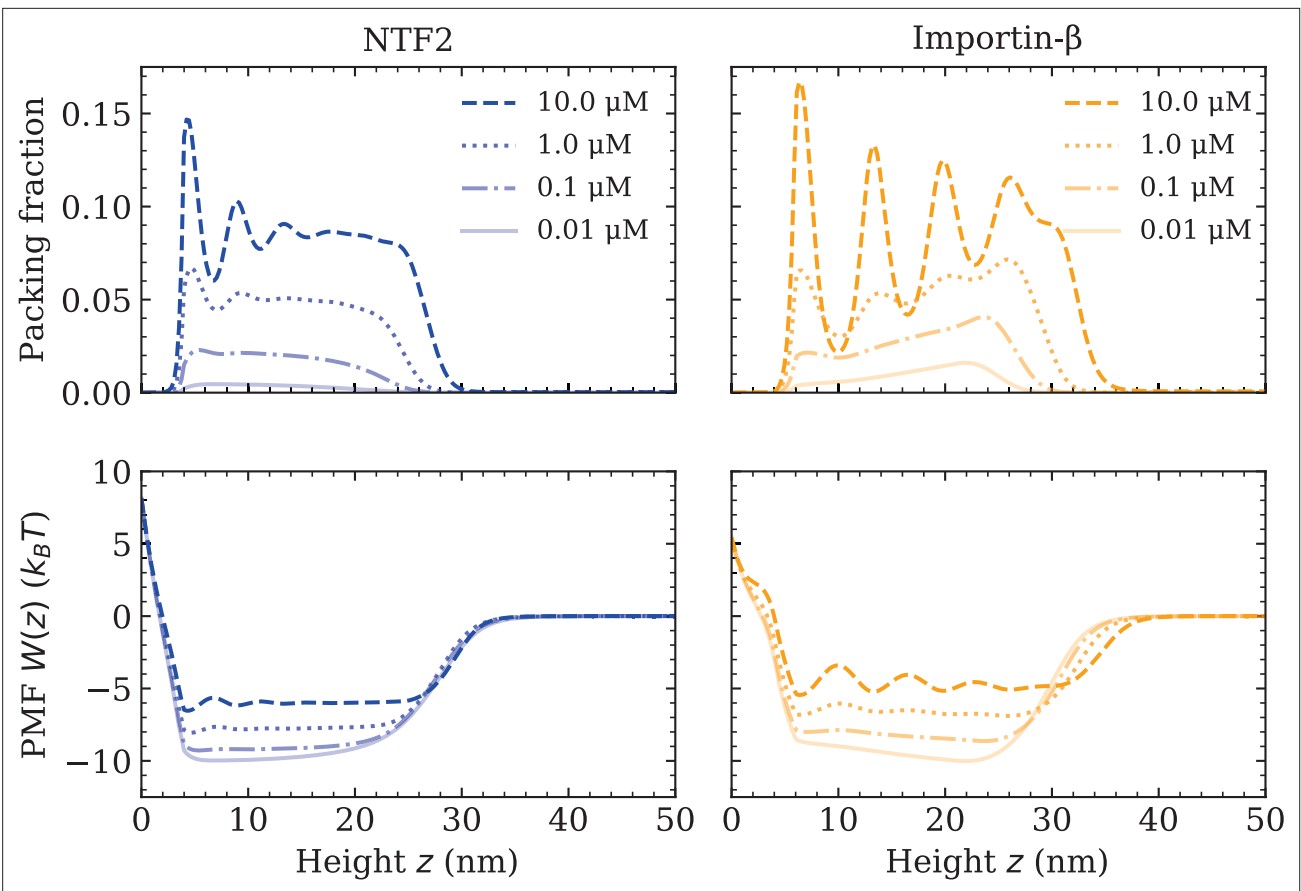

**Figure 2.** Increasing nuclear transport receptor (NTR) bulk concentration increases packing and filling up of the potential well within the Nsp1 film, for systems containing one type of NTR only. Equilibrium density functional theory packing fractions $\rho(z)d$, where $\rho(z)$ is the one-dimensional number density and $d$ is the particle diameter, as a function of the height $z$ above the flat surface for NTF2 (left) and Importin-$\beta$ (right), at various concentrations (top). Accompanying potentials of mean force $W(z)$ (bottom), for the same systems as on the top row.

The online version of this article includes the following figure supplement(s) for figure 2:

**Figure supplement 1.** The presence of NTF2 enhances the entry barrier for inert particles.

$$\text{NTR areal density in film} = \Gamma^{(i)}[\rho^{(i)}(z); \tau] = \frac{1}{A} \int_0^\tau \rho^{(i)}(z)\mathrm{d}z, \qquad (2)$$

where $\rho^{(i)}(z)$ ($i = \{1, 2\}$) is the number density of the NTRs. With only one free fitting parameter for each NTR (for the NTR-FG Nup interaction strength), the DFT binding isotherms are found to be in excellent agreement with experiment over three orders of magnitude in bulk NTR concentration (*Figure 1* [top]), as was previously accomplished by a similar DFT model (where polymers were attached to the base of a cylinder) in *Zahn et al., 2016*. For the here chosen interaction potential, the resulting parametrised cohesion strengths are $\epsilon_{\text{FG-NTF2}} = 2.4 \pm 0.1\ k_\text{B}T$ and $\epsilon_{\text{FG-Imp}\beta} = 2.3 \pm 0.1\ k_\text{B}T$ for NTF2 and Imp-$\beta$, respectively. It turns out that $\epsilon_{\text{FG-NTF2}} \approx \epsilon_{\text{FG-Imp}\beta}$ for the two (model) NTRs, despite the Imp-$\beta$ particle having an 1.5-fold larger excluded volume diameter as compared with the NTF2 particle. However, given the differences in diameters, and therefore a difference in the respective ranges of intermolecular interactions (see *equation 13*), we caution against directly comparing the cohesive properties of the two NTRs based on the cohesion strengths $\epsilon_{\text{FG-NTF2}}$ and $\epsilon_{\text{FG-Imp}\beta}$ alone. Of note, the concomitant film thicknesses from DFT – as a function of NTR concentration – are also in good agreement with experiment (*Figure 1* [bottom]).

At this point, all the essential interaction parameters $\epsilon_{\text{FG-FG}}$, $\epsilon_{\text{FG-NTF2}}$, and $\epsilon_{\text{FG-Imp}\beta}$ have been set by quantitative comparisons between DFT and experiment. Next, we investigate the effects of crowding of one type of NTR on the system. We quantify molecular crowding through two observables: (i) the packing fraction $\rho^{(i)}(z)\pi(d^{(i)})^3/6A$, where $\rho^{(i)}(z)$ is the effective 1D number density of a particular NTR

(labelled by $i$), and (ii) the PMF $W^{(i)}(z)$ of a particular NTR, in the presence of other NTRs and the FG Nups (see *Figure 2* and *equation 1*). For high crowding, one expects the packing fraction of a particular NTR to increase in magnitude and for the PMF to become more positive (with respect to the same system but with fewer NTRs), which is interpreted as a greater potential barrier (or, somewhat equivalently, a shallower potential well). We observe that both NTF2 and Imp-$\beta$ display higher levels of packing and higher-amplitude density oscillations within the Nsp1 film upon increasing their respective bulk concentrations (0.01, 0.1, 1.0, and 10.0 μM) (*Figure 2* [top]). The density oscillations arise from layering/ordering effects mainly caused by packing against a hard planar wall, where particles prefer to pack closer to a flat surface; the layering of hard-spheres next to a planar wall is a well-known phenomenon (*Patra, 1999*; *Roth et al., 2000*; *Deb et al., 2011*). As is expected, in both systems, the maximum observed packing fraction ($\gtrsim 0.15$) was located close to the surface (at 10 μM). For the here chosen NTR-particle sizes, it is expected that the packing fraction and PMF will be largely dictated by the interactions with the polymers and the crowding of other NTRs, with less significant effects arising from the particular implementation of the surface hardness. We note that the density oscillations for the Imp-$\beta$ particle show greater amplitudes as compared with the NTF2 particle (for the same concentration above the film), which is expected since the Imp-$\beta$ is larger in size and thus experiences more pronounced layering effects (*Padmanabhan et al., 2010*).

For both NTF2 and Imp-$\beta$, the PMFs decrease in magnitude (but remain negative within the bulk of the film) upon an increase in bulk NTR concentration (*Figure 2* [bottom]). Specifically, increasing the concentration in solution from 0.01 to 10.0 μM results in an approximate twofold decrease in the absolute value of the PMF ($|\Delta W(z)| \approx 4 - 5\,k_{\mathrm{B}}T$). The implication of this finding is that, at higher levels of packing in the film, it is relatively easier for bound NTRs to unbind from the polymer film, or, equivalently, less favourable for additional NTRs to enter the polymer film from the solution above it. This effect may primarily be attributed to the increased filling of space, i.e., molecular crowding, of the NTRs between the Nsp1 polymers. The results of *Figure 2* are particularly relevant to the NPC 'transport paradox', where fast transport (~1000 transport events per second) occurs in conjunction with strong – selective – binding. Whilst there are various explanations of the transport paradox (*Hoogenboom et al., 2021*), these results show how NTR crowding may facilitate the exit of NTRs from the NPC, noting that a decrease of $|\Delta W(z)| \approx 4 - 5\,k_{\mathrm{B}}T$ in PMF well depth would imply a ~100× faster rate for unbinding if we assume Arrhenius-like kinetics (*Figure 2* [bottom]).

Given the constant presence of NTRs in the NPC inner channel (*Lowe et al., 2015*; *Peters, 2009a*; *Peters, 2009b*; *Lim et al., 2015*; *Kim et al., 2018*; *Hoogenboom et al., 2021*), we wondered how varying the available bulk concentration of one type of NTR – in this case NTF2– would affect the entry barrier for inert particles as seen in our modelled FG Nup film (see *Figure 1—figure supplement 2*). As we increase the bulk concentration of NTF2 from 0.01 to 1.0 μM, we find that the amount of inert particles, with diameters in the range $d = 2.0 - 6.0$ nm, reduces within the FG Nup film, as compared to inert particles in a film with FG Nups only (see *Figure 2—figure supplement 1a*). The reduced amount of inert particles in the film is also quantified by an increase in the PMF, with the increase in PMF being larger for particles of greater size (see *Figure 2—figure supplement 1b*). This is in qualitative agreement with experimental observations of an enhancement of the NPC entry barrier to large inert particles through an increased presence of NTRs (*Kapinos et al., 2017*), and consistent with the notion that NTRs contribute to the integrity of the NPC transport barrier (*Peters, 2009a*; *Peters, 2009b*; *Lim et al., 2015*).

As a next step, we explore how the competition between NTRs may affect the binding, penetration, and distribution of NTRs in FG Nup assemblies. Specifically, we model the mixed crowding effects in a system containing the NTRs, NTF2 and Imp-$\beta$, in an Nsp1 polymer film (see *Figures 3 and 4* and their respective Figure supplements). As in the case with one type of NTR, we probe the packing fractions, PMFs $W^{(i)}(z)$, binding isotherms, and polymer film thickness, but this time keeping the amount of one NTR fixed at a physiologically relevant concentration (1 μM) (*Zahn et al., 2016*) whilst varying the concentration of the other NTR (*Figure 3a*). Upon increasing the bulk concentration of NTF2 from 0.01 to 10.0 μM while keeping the bulk concentration (in solution) of Imp-$\beta$ constant at 1 μM, the amount of bound Imp-$\beta$ dramatically drops and the remaining bound Imp-$\beta$ is redistributed towards the surface of the Nsp1 polymer film (*Figure 3a* [top] and *Figure 3—figure supplement 1*). Additionally, the density oscillations of Imp-$\beta$ within the film, which are evident at 0.01 μM of NTF2, smooth out upon increasing the amount of NTF2 to 0.1 μM. This shows that the presence of NTF2

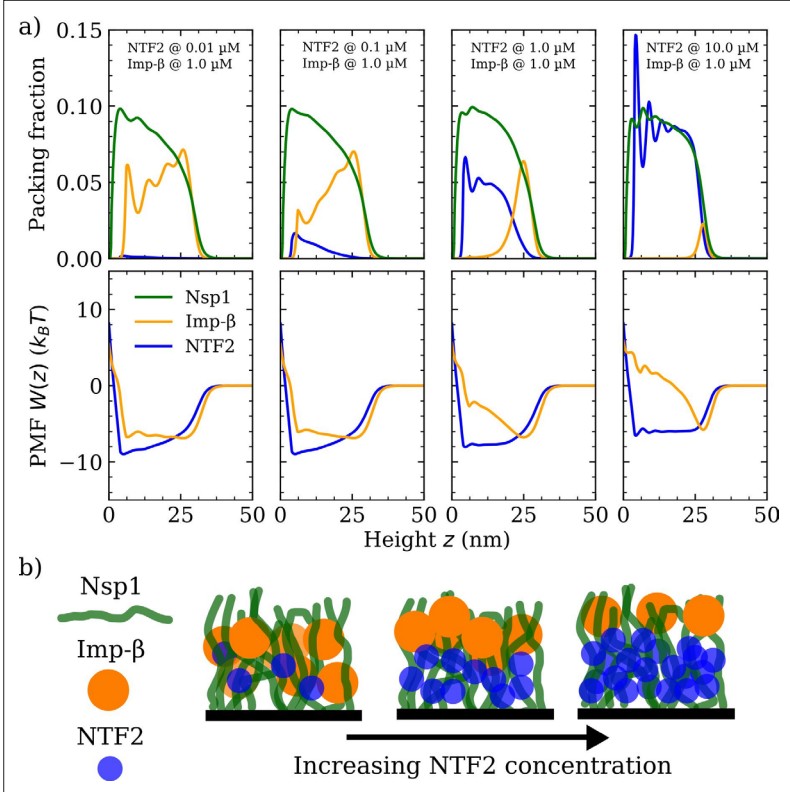

**Figure 3.** Phase separation in a ternary phenylalanine-glycine nucleoporins (FG Nup)-nuclear transport receptor polymer film assembly. (**a**) Density functional theory packing fractions and accompanying potentials of mean force (PMFs) for Nsp1 polymer films in the presence of NTF2 and Importin-$\beta$ (Imp-$\beta$). The concentration (in solution) of NTF2 is increased from 0.01 to 10.0 µM (left to right panels), whilst the concentration of Imp-$\beta$ is fixed at 1.0 µM. The cohesion strengths used here are {$\epsilon_{\text{FG-FG}} = 0.275\ k_BT, \epsilon_{\text{FG-NTF2}} = 2.4\ k_BT, \epsilon_{\text{FG-Imp}\beta} = 2.3\ k_BT$} for the Nsp1-Nsp1, Nsp1-NTF2, and Nsp1 - Imp-$\beta$ interactions, respectively. (**b**) Cartoon visualisation of the data from (**a**) depicting the increasing concentration of NTF2 pushing Imp-$\beta$ to the top of the Nsp1 layer, also resulting in significant expulsion of Imp-$\beta$ from the film into the bulk.

The online version of this article includes the following source code and figure supplement(s) for figure 3:

**Source code 1.** Simulation parameters for the classical density functional theory code.

**Figure supplement 1.** Nuclear transport receptor (NTR) binding isotherms and Nsp1 film thicknesses as a function of NTF2 concentration in solution, in the presence of 1 µM Importin-$\beta$ (Imp-β).

**Figure supplement 2.** Varying the cohesion between NTF2 molecules and phenylalanine-glycine nucleoporins (FG Nups) modulates the distribution of Importin-$\beta$ (Imp-β).

directly modulates the distribution of Imp-$\beta$ within the film. Interestingly, upon increasing NTF2 from 0.1 µM while keeping the bulk concentration of Imp-$\beta$ constant, we observe NTR phase separation: an NTF2-rich phase within the FG Nup film and an Imp-$\beta$-rich phase at the film surface. We can attribute this phase separation to the crowding of the NTRs, since by decreasing the affinity between NTF2 and the FG Nups, thereby decreasing the competitive advantage of NTF2 over Imp-$\beta$, we find an enhanced absorption of Imp-$\beta$ in the FG Nup polymer film (see ***Figure 3—figure supplement 2***).

When considering binary systems of hard-spheres with different diameters subject to packing between planar walls, ignoring any attractive interactions between them, one typically observes the larger particles packing closer to the wall, as compared with the smaller particles (***Padmanabhan et al., 2010***). This effect, as measured per unit area, is due to the overall system entropy loss being less when the larger particles pack closer to the surface, rather than the smaller ones. Here we observe the opposite effect, with the (smaller) NTF2 particles remaining closer to the grafting surface, which is qualitatively consistent with a theoretical study investigating a binary mixture of attractive particles, where the larger particles were excluded for a distance from a planar surface of up to twice

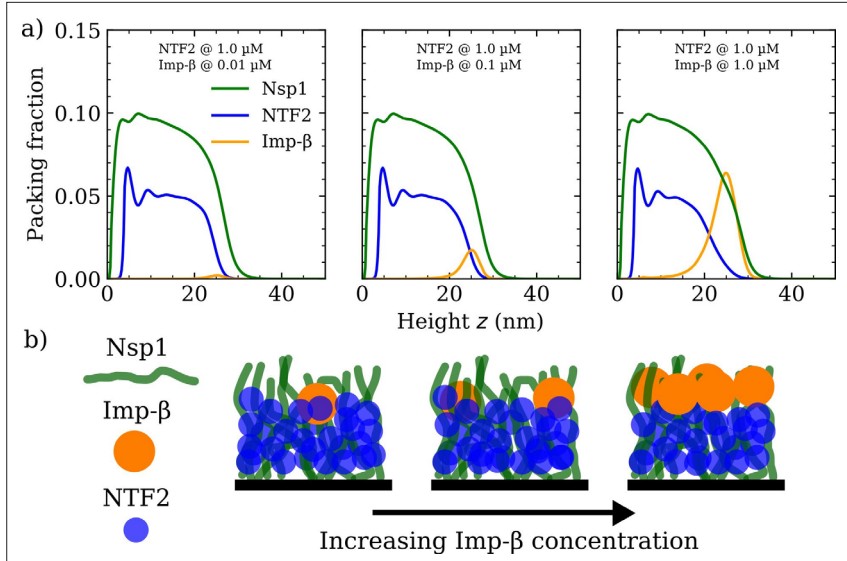

**Figure 4.** Increasing Importin-$\beta$ (Imp-$\beta$) concentration negligibly affects NTF2 in the Nsp1 film. (**a**) Density functional theory packing fractions against the height above the flat surface $z$ for Nsp1, NTF2, and Imp-$\beta$. The concentration of Imp-$\beta$ is increased from 0.01 to 10 µM (left to right panels) whilst the NTF2 concentration remains fixed at 1.0 µM. The last panel (furthest to the right) is the same as the second last panel in *Figure 3a*. (**b**) Cartoon illustration visualising the data in (**a**) depicting the undetectable change in the packing/morphology of the NTF2 in the presence of increasing Imp-$\beta$ molecules.

The online version of this article includes the following figure supplement(s) for figure 4:

**Figure supplement 1.** Nuclear transport receptor (NTR) binding isotherms and Nsp1 film thicknesses as a function of Importin-$\beta$ (Imp-β) concentration in solution.

the particle diameter (*Padmanabhan et al., 2010*). Here, however, we observe the depletion of the larger NTR (Imp-$\beta$) over much larger distances (in $z$) for high bulk concentrations of NTF2, apparently dictated by the polymer film thickness.

An intuitive explanation for the crowding-induced phase separation is that the smaller NTF2 competes more readily for binding sites (that are spread uniformly along the polymer in our model) deep within the film, closer to the grafting surface, without paying a substantial entropic penalty for rearranging the polymers. In contrast, closer to the film surface, the larger Imp-$\beta$ binds more readily, because of its overall stronger binding propensity, where the polymers are more diffuse (note $\epsilon_{\text{FG-NTF2}} \approx \epsilon_{\text{FG-Imp}\beta}$, spread over a larger particle surface for Imp-$\beta$). Indeed, the distribution of NTF2 in the film largely follows the polymer density as a function of distance from the grafting surface, indicating that with its smaller size, NTF2 benefits more from the enhanced concentration of polymer beads (and therewith of binding sites) without having to pay a substantial entropic cost (as for Imp-$\beta$) for penetrating the polymer film. A similar phenomenon has also been observed in a binary system of small and large positively charged particles soaked in a solution of anions, with a negatively charged surface at the bottom (*Fang and Szleifer, 2003*).

Throughout the changes in incorporation of NTF2, the density of the polymers did not show noticeable changes. The modulation of Imp-$\beta$ via changes in NTF2 numbers is also articulated in terms of the PMF $W(z)$, where the Imp-$\beta$ PMF is an approximate square well at 0.01 µM of NTF2, but for higher NTF2 concentrations gradually transforms into a pronounced and sharper well located at $z \approx 25.0$ nm, i.e., at the surface of the film, with the formation of a barrier to enter the rest of the film (*Figure 3a* [bottom]).

We verified if similar effects resulted when increasing the concentration of Imp-$\beta$ for a given, constant, NTF2 concentration set at 1.0 µM (see *Figure 4a* and *Figure 4—figure supplement 1*). We observe no significant change to the Nsp1 or NTF2 packing factions (including the PMF and binding isotherm) upon increasing the concentration of Imp-$\beta$ in solution from 0.01 to 1.0 µM (see also *Figure 4b*). We have not explored high bulk concentrations (>1 µM) of Imp-$\beta$, as these yield highly oscillatory packing fractions and therewith are computationally more challenging in our DFT model.

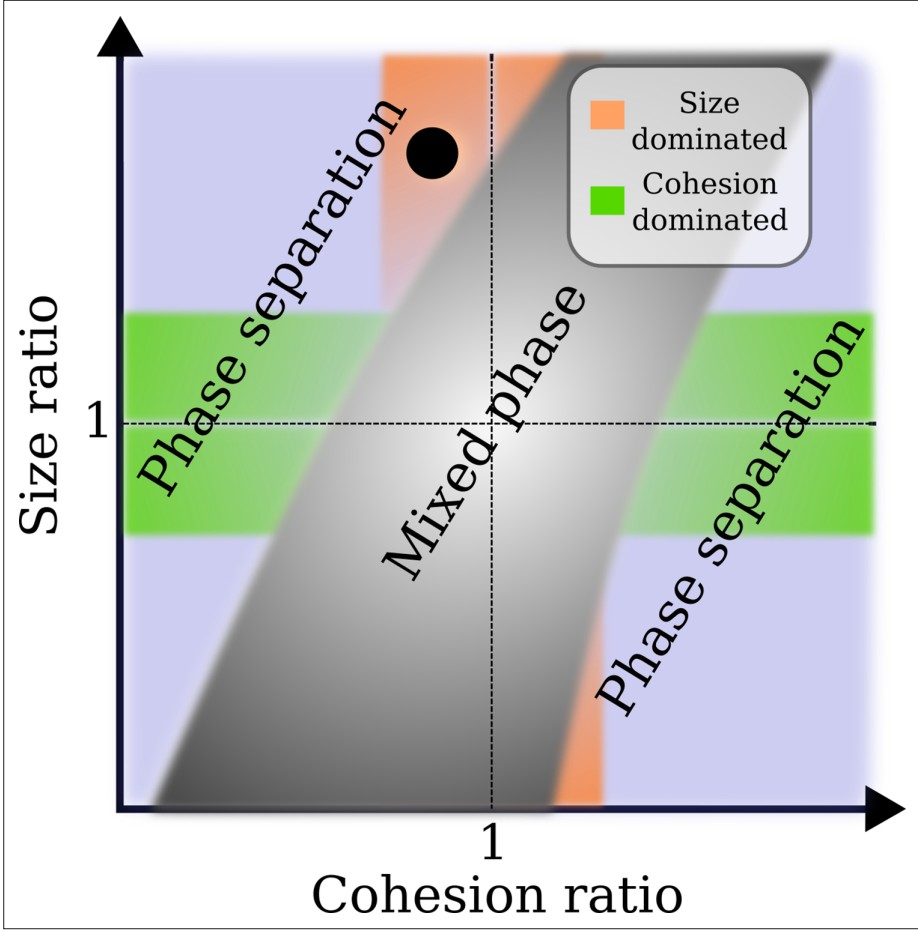

**Figure 5.** Approximate phase diagram for the crowding of two different types of nuclear transport receptor (NTR)-like particles in an Nsp1 film, based on simulations for particle sizes $d^{(1,2)}$ between 2 and 6 nm, and their affinities $\epsilon^{(13,23)}$ (cf., $\epsilon_{\text{FG-NTF2}}$ and $\epsilon_{\text{FG-Imp}\beta}$) to Nsp1 between ~0 and ~10 $k_{\text{B}}T$. Plotted as a linear function of both the size ratio and cohesion (or affinity) ratio, the phase diagram shows a mixed phase (grey), where the density profiles of the NTR-like particles in the Nsp1 film have substantial overlap, and a phase separated state (not grey), where the profiles of the NTRs are sufficiently separated resulting in an interface. In the orange region the cohesion ratio of the NTRs is close to unity whereas the size ratio is far from unity, indicating that the phase separation results from the difference in NTR size. The green region is the exact opposite of the orange region: the cohesion ratio of the NTRs is far from unity whereas the size ratio is close to unity, indicating that the phase separation results from the difference in NTR cohesion. The blue region encompasses the cases where both the size and cohesion ratios are far from unity. The black circle represents the phase separation of modelled Importin-$\beta$ (Imp-$\beta$) and NTF2 ($d_{\text{Imp-}\beta}/d_{\text{NTF2}} = 1.5$ and $\epsilon_{\text{FG-Imp}\beta}/\epsilon_{\text{FG-NTF2}} \approx 0.96$) as found in this work.

The online version of this article includes the following figure supplement(s) for figure 5:

**Figure supplement 1.** Relative density plots for selected points in the phase diagram.

**Figure supplement 2.** Modelled nuclear transport receptor (NTRs) particles of diameters 2 nm and 4 nm hardly affect each other's distribution within the film.

However, we expect that further incorporation of Imp-$\beta$ would eventually change the distribution of NTF2 in the film.

To further establish the factors that define NTR crowding and competition in our model Nsp1 system with two types of NTR-like particles, we explored the phase diagram as a function of the respective affinities $\epsilon^{(13)}, \epsilon^{(23)}$ of the two NTRs to Nsp1, and as a function of their respective sizes $d^{(1)}, d^{(2)}$. Varying the particle sizes between 2 and 6 nm, and the affinities between ~0 and ~10 $k_{\text{B}}T$ (**Figure 3—figure supplement 2**, **Figure 5—figure supplement 1**, and **Figure 5—figure supplement 2**), we find mixed and segregated phases of the NTR-like particles in the Nsp1 assembly (**Figure 5**). Not surprisingly, full mixing between the two types of particles occurs when $d^{(2)}/d^{(1)} = 1$ and $\epsilon^{(23)}/\epsilon^{(13)} = 1$,

i.e., when the particles are identical. Within the explored parameter range, phase separation occurs when these respective ratios become sufficiently small or large. Physically, the observed behaviour can be readily understood by noting that the entropic cost of penetrating the film increases with particle size (such that, e.g. this cost is larger for Imp-$\beta$ than for NTF2), and that the energy gain of penetrating the film increases with the particle affinity to Nsp1. At the cross-over between these two competing effects, particles will favour binding at the surface of the FG Nup assembly. Part of this is trivial, e.g., small enough particles would readily penetrate the film, as their presence does not significantly affect the entropy related to the various possible polymer configurations; consequently, substantially larger disparities in the respective FG Nup-NTR cohesion strengths are required to phase-separate systems with NTR-like particles sized ≲4 nm (*Figure 5—figure supplement 2*). The intriguing aspects here are that the polymer-related entropic cost of NTR binding just about kicks in when NTR sizes are a few nanometres (as is true for physiological NTRs), which is at least fourfold larger than the FG Nup persistence length and bead size that are of the order of 1 nm. Additionally, the respective sizes and Nsp1 affinities of NTF2 and Imp-$\beta$ are such that inter-NTR competition readily drives the system from mixed to phase separated states.

## Discussion

NTRs are crucial to the function of the NPC and are increasingly recognised to be continuously present in the FG Nup assembly that dictates transport selectivity through the NPC in vivo (*Lowe et al., 2015*; *Peters, 2009a*; *Peters, 2009b*; *Lim et al., 2015*; *Kim et al., 2018*; *Hoogenboom et al., 2021*), yet their configuration inside the NPC remains poorly characterised. In this study, we identified the physical behaviour that can occur when different types of NTRs partition in an FG Nup assembly that mimics the NPC transport channel by its FG Nup density, by being confined to an assembly of only few tens of nm thick, and by binding or excluding NTRs and inert particles in a manner consistent with NPC transport functionality.

Specifically, we have made quantitative and testable predictions regarding the effects of mixed crowding on the binding of different NTRs to FG Nup planar assemblies. Our results are based on a minimal coarse-grained model implemented in a mean-field approach, using classical DFT, similar to our previous model for the binding of a single type of NTR to FG Nup assemblies (*Zahn et al., 2016*), that treats FG Nups as sticky and flexible homopolymers and NTRs as isotropic and cohesive spheres, with excluded volume interactions between all components based on their known sizes. The model here includes three free interaction parameters, corresponding to the cohesive FG Nup-FG Nup and NTR-FG Nup (for two types of NTR) interactions, i.e., no cohesive interactions between NTRs themselves. These cohesion parameters were parametrised using experimental data for Nsp1 film assemblies and binding thereto of one type of NTR (NTF2 or Imp-$\beta$).

In *Zahn et al., 2016*, the FG Nup assembly was modelled via an effectively 2D approach of polymers grafted at the bottom of a wide cylinder with rotational symmetry. Here, we have instead imposed translational invariance across the anchoring surface of the FG Nups therefore reducing the problem to an effectively 1D one, where the distribution of particles depends only on the height $z$ above the surface. Apart from changes in the effective interaction parameters that are largely due to the different choice of interaction potential here, the results are fully consistent with (*Zahn et al., 2016*). Compared with this previous work, however, we have here studied the effects of crowding on the NTR binding to the FG Nup assembly, and more specifically considered the case of simultaneous binding by two different types of NTRs, NTF2 and Imp-$\beta$, to a planar assembly of Nsp1.

Based on the parametrised model, we have shown that increased crowding of one type of NTR results in a shallower potential well within the film, implying that individual NTRs will have a smaller potential barrier to leaving the film in the presence of more NTRs. The origin of this effect is due to an interplay between the further occupation of volume within the film (entropic) and the increased competition for binding sites. This result has important implications for the NPC: when there is a large influx of material into the channel from either the cytoplasm or nucleoplasm, the exit of said material should be faster since the increased crowding effects will reduce the free energy barrier – thus increasing the likelihood – to leave the pore, with a predicted increase in unbinding rates of two orders of magnitude in the concentration range explored here. While we note that there are multiple factors involved in determining transport speed (*Hoogenboom et al., 2021*), this scenario highlights the importance of NTRs as an essential component in the NPC transport barrier (*Lim et al., 2015*)

and, specifically, implies that the NPC could perform more efficiently and faster with higher numbers of NTRs present in its inner channel, as has indeed been observed in experiments with Imp-$\beta$ (**Yang and Musser, 2006**).

In addition, we have found that with increased incorporation of NTF2 within the FG Nup film, the amount of absorbed Imp-$\beta$ was reduced and its distribution within the film was changed. Typically, the smaller NTR binds closer to the bottom of the film (where the polymer packing is higher) and the larger NTR is more likely to bind to the top of the film (where the polymer packing is lower). This observation can be attributed to the smaller entropic penalty incurred when smaller NTRs (here: NTF2) penetrate the FG Nup assembly, compared with larger NTRs (here: Imp-$\beta$). The intriguing aspect here is that within a physiologically relevant parameter range, NTF2 modulates the absorption and penetration of Imp-$\beta$ in the FG Nup assembly, driving the system to phase separate, resulting in spatially segregated regions that are enriched in the two respective NTRs.

Generally, the here discussed competitive binding phenomenon bears similarity to the observation that NTR binding to FG Nups may be modulated by more weakly, non-specifically binding proteins, provided that the latter are present at sufficiently high concentrations, as in fact validated by experiments on planar FG Nup assemblies (**Lennon et al., 2021**). Physically, the differential binding of smaller and larger NTRs can be compared with that of smaller positively charged particles packing closer to a negatively charged surface, with the larger positively charged particles sitting on top of the smaller ones, due to the higher entropic costs of packing larger particles close to an attractive surface (**Fang and Szleifer, 2003**). The case here is different, however, in that the polymer behaviour of the FG Nups further enhances differences in entropic cost for absorption of differently sized particles.

Given the simplified nature of our NTR representation as homogeneous spheres, we cannot draw definite conclusions about how the observed NTR behaviour may depend on any cargo associated with the NTRs. However, given the general trends in the phase diagram (**Figure 5**), we speculate that the binding of differently sized cargoes may lead to further magnified phase separation between NTR-cargo complexes in FG Nup assemblies. For similar practical considerations, the details of multivalency/avidity at the NTR surface are beyond the scope of this work, but based on our previous work on isolated FG Nups (**Davis et al., 2021**), we expect that such multivalency/avidity will play a large role in the kinetics of layering and entry/exit of the FG Nup assembly; and note that such multivalency also presents subtle challenges with regard to the thermodynamics of the selective barrier (**Matsuda and Mofrad, 2021**).

It is important to note that the observed height-dependent phase separation, as predicted in our minimal 1D model (assuming symmetry parallel to the anchoring surface), may not be the only way for NTRs to spatially segregate. Considering the pore geometry of the NPC, it would be interesting to extend the model developed here to two or three dimensions, relaxing the lateral symmetry assumption (**Osmanović et al., 2013b**), and to consider how radial gradients in polymer/FG Nup density – as, e.g., in **Osmanovic et al., 2012**; **Perez Sirkin et al., 2021** – could facilitate phase separation along the radial axis of the NPC channel. An immediate consequence of this is that transport pathways through the NPC are most likely dependent on the type of NTR, with potentially separate transport pathways mediated and modulated by different NTRs.

Given that there is a stable population of Imp-$\beta$ in the NPC barrier and given that changes to this population affect the selective properties of the NPC (**Lowe et al., 2015**), our results suggest that NTR crowding plays a substantial role in determining the performance of the NPC barrier. Additionally, the observation of a phase-separated state between two distinct NTRs has implications on how the NPC maintains high-throughput transport despite high NTR densities. Consistent with experimental observations on NPCs (**Lowe et al., 2015**), Imp-$\beta$ is found to occupy regions of lower FG Nup density (as shown here in planar FG Nup assemblies), where our results here demonstrate that such a distribution of Imp-$\beta$ can at least in part be attributed to competitive binding of other, smaller NTRs to regions of higher FG Nup density. Finally, our results enrich reduction-of-dimensionality and 'Kap-centric' perspectives to the NPC transport barrier (**Peters, 2005**; **Wagner et al., 2015**), in that small NTRs preferentially occupy regions in the NPC inner channel that are denser in FG motifs whilst larger NTRs preferentially occupy more dilute regions, leading to spatially segregated transport pathways, and/or with the specific transport associated with one type of NTR being modulated by another type of NTR, thus providing additional levels of control for selective transport through the NPC.

## Methods

We use classical DFT, an equilibrium mean field theory developed in previous works (*Osmanović et al., 2013b*; *Zahn et al., 2016*; *Davis et al., 2020*), to model the FG Nup-NTR planar film assembly consisting of NTF2, Imp-$\beta$, and Nsp1.

We first build a rather generic physical model consisting of a ternary mixture, i.e., a $\nu$-component system with $\nu = 3$, containing two types of free particles denoted by $i = 1, 2$ (which will describe the NTRs) and one type of polymer labelled as $i = 3$ (that will describe the anchored FG Nup Nsp1). Here, we do not explicitly describe the small-scale solvent molecules, as the dominant interactions come from the NTRs and the FG Nups. However, the solvent medium is implicitly accounted for when parametrising the intramolecular and intermolecular interactions between FG Nups and the interactions between NTRs and FG Nups (*Zahn et al., 2016*; *Hoogenboom et al., 2021*). In this system, the numbers of the two different types of free particles (components $i = 1, 2$) are given as $N^{(i)}$, diameters are $d^{(i)}$, and chemical potentials are $\mu^{(i)}$. In addition to the free particles, there are $N^{(i=3)} = 260$ flexible homopolymers each consisting of $M = 300$ beads, where each bead has a diameter of $d^{(3)} = 0.76$ nm (corresponding to two amino acids per bead). This choice of $M$ and $d^{(3)}$ produces the approximate contour and persistence length of an Nsp1 FG domain (*Lim et al., 2006*; *Zahn et al., 2016*; *Hayama et al., 2019*; *Davis et al., 2020*). The polymers are attached uniformly to a flat surface of area $A = 88.62 \times 88.62$ nm$^2$, resulting in an attachment/grafting density of $\approx 3.3$ polymers/100 nm$^2$, which is in line with the densities in the native NPC and in in vitro experiments (*Zahn et al., 2016*; *Davis et al., 2020*). For simplicity, it is assumed that the system is translationally symmetric along the directions parallel to the grafting surface, therefore, after integrating out the $x - y$ plane the density of component , $\rho^{(i)}(\mathbf{r})$ can be transformed into a function of $z$ (height above the anchoring surface) only:

$$\rho^{(i)}(z) = \int \int dx dy \rho^{(i)}(\mathbf{r}), \tag{3}$$

$$\rho^{(i)}(z) = A \rho^{(i)}(\mathbf{r}), \tag{4}$$

resulting in an effective 1D DFT (*Davis et al., 2020*).

Furthermore, the grand potential free energy functional $\Omega$, that provides a complete thermodynamic description of the entire system, is approximated as a sum of terms.

$$\begin{aligned} \Omega = \quad &\mathcal{F}_{\text{ideal-gas}} + \mathcal{F}_{\text{ideal-polymer}} + \mathcal{F}_{\text{mean-field}} + \mathcal{F}_{\text{external}} + \\ &\mathcal{F}_{\text{exchange}} + \mathcal{F}_{\text{cohesion}} + \mathcal{F}_{\text{hard-sphere}}. \end{aligned} \tag{5}$$

The first term is the free energy of a set of two types of ideal gas (for the two types of NTRs) given as

$$\mathcal{F}_{\text{ideal-gas}} = \beta^{-1} \sum_{i=1}^{2} \int dz \rho^{(i)}(z) \left( \ln((\lambda^{(i)})^3 A^{-1} \rho^{(i)}(z)) - 1 \right), \tag{6}$$

where $\beta = 1/k_B T$ ($k_B$ is Boltzmann's constant) and $\lambda^{(i)}$ is the appropriate thermal de Broglie wavelength for component $i$. The second term describes the ideal polymer free energy (in the presence of a mean field $w(z)$) and is given as (*Fredrickson, 2005*)

$$\mathcal{F}_{\text{ideal-polymer}} = -N^{(3)} \beta^{-1} \ln(Z_c[w(z)]) \tag{7}$$

where the canonical partition function $Z_c$ is

$$Z_c[w(z)] = \frac{A^N}{N!(\lambda^{(3)})^{3N}} \int dz^N \exp\left[ -\beta \sum_{m=1}^{N^{(3)}} \sum_{i=1}^{M} h_m(z_{i+1}, z_i) - \sum_{m=1}^{N^{(3)}} \sum_{i=1}^{M} w(z_{im}) \right], \tag{8}$$

where $N = M \times N^{(3)}$, $\int dz^N \equiv \int \prod_{i=1}^{N} dz_i$, $h_m(z_{i+1}, z_i)$ is an energy function that imposes a rigid bond length between directly connected beads (pairs $\{i + 1, i\}$ for $1 \leq i \leq M$) on polymer $m$, and $w(z_{im})$ is the mean field contribution of the $i$th bead belonging to polymer $m$ (*Davis et al., 2020*). The third term, $\mathcal{F}_{\text{mean-field}}$, is the additional term that mathematically arises from introducing a mean field description of the affinities between the polymers (*Osmanovic et al., 2012*) and is given as

$$\mathcal{F}_{\text{mean-field}} = -\beta^{-1} \int dz w(z) \rho^{(3)}(z). \tag{9}$$

The fourth term, $\mathcal{F}_{\text{external}}$, accounts for the external potential imposing the hardness of the anchoring surface and is determined through

$$\mathcal{F}_{\text{external}} = \sum_{i=1}^{3} \int dz \rho^{(i)}(z) V_{ext}^{(i)}(z), \tag{10}$$

where $V_{ext}^{(i)}(z)$ is a repulsive external potential energy function taking a Weeks-Chandler-Anderson form

$$V_{ext}^{(i)}(z) = \begin{cases} 4\epsilon_{ext} \left[ \left( \frac{\sigma^{(i)}}{z} \right)^{12} - \left( \frac{\sigma^{(i)}}{z} \right)^{6} \right] + \epsilon_{ext}, & z < d^{(i)}, \\ 0, & z \geq d^{(i)}, \end{cases} \tag{11}$$

where $d^{(i)}$ is the diameter of the constituent particle for component , $\epsilon_{ext} = 20$ is the maximum energy barrier of the wall (chosen sufficiently high so that the number density of all components is zero at and below the surface), and $\sigma^{(i)} = 2^{-1/6} d^{(i)}$. The fifth term, $\mathcal{F}_{\text{exchange}}$, imposes an exchange of the NTRs with an external reservoir, leading to a grand canonical ensemble, and is written as

$$\mathcal{F}_{\text{exchange}} = -\sum_{i=1}^{2} \mu^{(i)} \int dz \rho^{(i)}(z). \tag{12}$$

Consistent with our previous work (**Davis et al., 2021**), the intramolecular and intermolecular cohesive interactions are based upon the Morse potential (in three dimensions)

$$u_{3D}^{(ij)}(r) = \epsilon^{(ij)} \left( e^{-2\alpha(r-d^{(ij)})} - 2e^{-\alpha(r-d^{(ij)})} \right), \tag{13}$$

where $r$ is the distance between two particles of type  and type $j$, $\epsilon^{(ij)}$ is the cohesion strength, $\alpha = 6.0$ nm⁻¹ is an inverse decay length of the pair potential, and $d^{(ij)} = \frac{1}{2}(d^{(i)} + d^{(j)})$. The potential in **equation (13)**, valid in three spatial dimensions, is then integrated over the plane, making it henceforth only dependent on $z$, and shifted such that $u^{(ij)}(z) = 0$ for $z \geq 2d^{(ij)}$ so as to keep the cohesive interactions short ranged. Thus we can now define the sixth term, $\mathcal{F}_{\text{cohesion}}$, as the free energy contribution from intermolecular and intramolecular attractive ('cohesive') interactions determined by

$$\begin{aligned} \mathcal{F}_{\text{cohesion}} &= \frac{1}{2} \sum_{i=1}^{3} \int \int \rho^{(i)}(z) \rho^{(i)}(z') u^{(ii)}(z - z') dz dz', \\ &+ \sum_{i=1}^{2} \sum_{j=i+1}^{3} \int \int \rho^{(i)}(z) \rho^{(j)}(z') u^{(ij)}(z - z') dz dz'. \end{aligned} \tag{14}$$

The final term, $\mathcal{F}_{\text{hard-sphere}}$, accounts for the intermolecular and intramolecular excluded volume interactions, including imposing polymer chain connectivity, and is given by

$$\mathcal{F}_{\text{hard-sphere}} = \int dz \left( \phi^{WB} \left( n_\alpha(z), \boldsymbol{n}_\alpha(z) \right) + \phi^{CH} \left( n_\alpha^{(3)}(z), \boldsymbol{n}_\alpha(z) \right) \right), \tag{15}$$

where $\phi^{WB}$ and $\phi^{CH}$ are the White Bear (hard-sphere) (**Roth et al., 2002**) and chain connectivity functionals (**Yu and Wu, 2002**) given by the equations

$$\phi^{WB} = -n_0 \ln(1 - n_3) + \frac{n_1 n_2 - \boldsymbol{n}_1 \cdot \boldsymbol{n}_2}{1 - n_3} + (n_2^3 - 3n_2 \boldsymbol{n}_2^2) \frac{n_1 + (1 - n_3)^2 \ln(1 - n_3)}{36\pi n_3^2 (1 - n_3)^2}, \tag{16a}$$

$$\phi^{CH} = \left( \frac{1 - M}{M} \right) n_0 \left( 1 - \frac{\boldsymbol{n}_2^2}{n_2^2} \right) \ln \left( \frac{1}{1 - n_3} + \frac{n_2 R (1 - \frac{\boldsymbol{n}_2^2}{n_2^2})}{2(1 - n_3)^2} + \frac{n_1 R^2 (1 - \frac{\boldsymbol{n}_2^2}{n_2^2})}{18(1 - n_3)^3} \right), \tag{16b}$$

where $n_\alpha(z; \{\rho^{(i)}\})$ and $\boldsymbol{n}_\alpha(z; \{\rho^{(i)}\})$ are, respectively, the 1D scalar and vectorial weighted densities and $R$ is the radius of a polymer bead (see **Roth, 2010**). Essentially, the White Bear functional removes – from the free energy – contributions from system configurations with particle overlaps, which is the

definition of a hard-sphere system. The hard-sphere functional is built using a standard formalism known as fundamental measure theory, attributed to Rosenfeld, which begins with a decomposition of the Mayer function into weight functions that contain geometrical information of spherical particles (see **Rosenfeld, 1989**; **Roth et al., 2002**; **Roth, 2010** for details). The chain-connectivity functional is built in a similar manner, but removes from the free energy contributions from system configurations that violate how consecutive beads on a chain should be connected, as specified by the particular polymer model (see **Yu and Wu, 2002** for more details).

Thus, the dimensionless grand potential can be written more explicitly as

$$
\begin{aligned}
\beta\Omega = \quad & \sum_{i=1}^{2} \int \mathrm{d}z \rho^{(i)}(z)(\ln((\lambda^{(i)})^3 A^{-1}\rho^{(i)}(z)) - 1) - N^{(3)}\ln(Z_C[w]) - \int \mathrm{d}z w(z)\rho^{(3)}(z) \\
& + \beta \sum_{i=1}^{2}\int \mathrm{d}z \rho^{(i)}(z)\left(V_{ext}^{(i)}(z) - \mu^{(i)}\right) + \beta \int \mathrm{d}z \rho^{(3)}(z)V_{ext}^{(3)}(z) \\
& + \beta \frac{1}{2}\sum_{i=1}^{3}\int\int \rho^{(i)}(z)\rho^{(i)}(z')u^{(ii)}(z-z')\mathrm{d}z\mathrm{d}z' \\
& + \beta \sum_{i=1}^{2}\sum_{j=i+1}^{3}\int\int \rho^{(i)}(z)\rho^{(j)}(z')u^{(ij)}(z-z')\mathrm{d}z\mathrm{d}z' \\
& + \int \mathrm{d}z \left(\phi^{WB}\left(n_\alpha(z), \boldsymbol{n}_\alpha(z)\right) + \phi^{CH}\left(n_\alpha^{(3)}(z), \boldsymbol{n}_\alpha(z)\right)\right)
\end{aligned}
\tag{17}
$$

To find the set of density distributions – for the particles and polymer – and the polymer mean field in the equilibrium state, the following equations must be solved self-consistently:

$$
\frac{\beta\delta\Omega}{\delta w} = \int \mathrm{d}z \left[-w(z) + c^{(3)}(z) + \beta\sum_{i=1}^{3}\int \rho^{(i)}(z)u^{(i3)}(z-z')\mathrm{d}z' + \beta V_{ext}^{(3)}(z)\right]\frac{\delta\rho^{(3)}[w(z)]}{\delta w(z'')} = 0,
\tag{18a}
$$

$$
\frac{\beta\delta\Omega}{\delta\rho^{(i)}} = c^{(i)}(z) + \ln(\lambda^{(i)}\rho^{(i)}(z)) + \beta\sum_{j=1}^{3}\int \rho^{(j)}(z)u^{(ij)}(z-z')\mathrm{d}z' + \beta\left(V_{ext}^{(i)}(z) - \mu^{(i)}\right) = 0, \quad i = 1,2
\tag{18b}
$$

$\frac{\delta}{\delta x}$ where the notation represents a functional derivative with respect to and is the one-body direct correlation function given by

$$
c^{(i)}(z) = \beta\frac{\delta\mathcal{F}_{\text{hard-sphere}}[\rho^{(i)}]}{\delta\rho^{(i)}(z)} = \sum_\alpha \int \mathrm{d}z'\frac{\delta\phi^{WB}}{\delta n_\alpha^{(i)}}\frac{\delta n_\alpha^{(i)}(z')}{\delta\rho^{(i)}(z)}, \quad i = 1,2,
\tag{19a}
$$

$$
c^{(3)}(z) = \beta\frac{\delta\mathcal{F}_{\text{hard-sphere}}[\rho^{(3)}]}{\delta\rho^{(3)}(z)} = \sum_\alpha \int \mathrm{d}z'\frac{\delta(\phi^{WB}+\phi^{CH})}{\delta n_\alpha^{(3)}}\frac{\delta n_\alpha^{(3)}(z')}{\delta\rho^{(3)}(z)}.
\tag{19b}
$$

For the free particles one can decompose the chemical potential into two terms

$$
\mu^{(i)} = -\beta^{-1}\ln\left(\frac{1}{(\lambda^{(i)})^3 A^{-1}\rho_{\text{bulk}}^{(i)}}\right) + \mu_{exc}^{(i)}, \quad i = \{1,2\},
\tag{20}
$$

where $\rho_{\text{bulk}}^{(i)}$ is the bulk density of the free particles of component $i$ and $\mu_{exc}^{(i)}$ is the excess chemical potential due to the intermolecular and intramolecular interactions. One can solve **equations (18)** self-consistently by invoking a fictitious time variable $t$, where the solutions are found through an iterative procedure. This is expressed by the following

$$
\frac{\partial w(z)}{\partial t} = -w(z) + c^{(3)}(z) + \beta\sum_{i=1}^{3}\int p^{(i)}(z)u^{(13)}(z-z')\mathrm{d}z' + \beta V_{ext}^{(3)}(z),
\tag{21a}
$$

$$
\begin{aligned}
\frac{\partial\rho^{(i)}(z)}{\partial t} \quad = \quad & -\rho^{(i)}(z) + \rho_{\text{bulk}}^{(i)}\times \\
& \exp\left(\beta\mu_{exc}^{(i)} + c^{(i)}(z) - \beta\sum_{j=1}^{3}\int \rho^{(j)}(z)u^{(ij)}(z-z')\mathrm{d}z' - \beta V_{ext}^{(i)}(z)\right), \quad i = 1,2.
\end{aligned}
\tag{21b}
$$

Finally, discretising space into slices of thickness and discretising fictitious time then yields the resulting discrete update rules which are solved numerically

$$
w_{n+1}(z_j) = w_n(z_j) + \Delta t\left(-w_n(z_j) + c^{(3)}(z_j) + \beta\sum_{i=1}^{3}\sum_{k=0}^{L}\rho^{(i)}(z_k)u^{(i3)}(z_k - z_j')\Delta z + \beta V_{ext}^{(3)}(z_j)\right),
\tag{22a}
$$

$$\begin{aligned}\rho_{n+1}^{(i)}(z_j) \quad &= \rho_n^{(i)}(z_j) - \Delta t \rho^{(i)}(z_j) + \Delta t \rho_{\text{bulk}}^{(i)} \\ &\exp\left(\beta\mu_{exc}^{(i)} + c^{(i)}(z_j) - \beta \sum_{m=1}^{3}\sum_{k=0}^{L} \rho^{(m)}(z_k)u^{(im)}(z_k - z_j)\Delta z - \beta V_{ext}^{(i)}(z_j)\right),\end{aligned} \quad (22b)$$

where $z_k$ is the – midpoint – height above the surface of the spatial slice $k$, $n$ labels discrete time, and in the last equation $i = 1, 2$. The simulation parameters used here were $L = 1024$, $\Delta z = 0.117$ nm (with $z_0 = 0.0585$), and $\Delta t = 0.002$.

We note that, for the free particles $i = 1, 2$, an excess chemical potential ($\mu_{exc}^{(i)}$ for $i = 1, 2$) is referenced to a zeroed chemical potential $\beta\mu_{exc}^{(i)} = 0$ that results in a free particle bulk concentration of 10 µM. Changing the excess chemical potential to $\beta\mu_{exc}^{(i)} = \pm 2$ results in an order of magnitude increase (for +2) or decrease (for –2) of the concentration in solution (see also *Osmanović et al., 2013b*).

## Acknowledgements

We thank Dino Osmanović, Anton Zilman, and Anđela Šarić for discussions. We thank Ralf Richter for providing feedback on the manuscript. LKD acknowledges the biophysics research computing cluster at UCL that was used to perform the simulations and analysis. This work was funded by the UK Engineering and Physical Sciences Research Council (EP/L504889/1, LKD and BWH).

## Additional information

### Funding

| Funder | Grant reference number | Author |
|---|---|---|
| Engineering and Physical Sciences Research Council | EP/L504889/1 | Luke K Davis<br>Bart W Hoogenboom |

The funders had no role in study design, data collection and interpretation, or the decision to submit the work for publication.

### Author contributions

Luke K Davis, Conceptualization, Data curation, Formal analysis, Investigation, Methodology, Project administration, Software, Validation, Visualization, Writing – original draft, Writing – review and editing; Ian J Ford, Methodology, Supervision, Validation, Writing – review and editing; Bart W Hoogenboom, Conceptualization, Funding acquisition, Methodology, Project administration, Resources, Supervision, Validation, Writing – review and editing

### Author ORCIDs

Luke K Davis (iD) http://orcid.org/0000-0003-4487-4159
Bart W Hoogenboom (iD) http://orcid.org/0000-0002-8882-4324

### Decision letter and Author response

Decision letter https://doi.org/10.7554/eLife.72627.sa1
Author response https://doi.org/10.7554/eLife.72627.sa2

## Additional files

### Supplementary files

• Transparent reporting form

### Data availability

The source code used to generate all the simulation data in this manuscript is available on the Github repository: https://github.com/patherlkd/DFT-polymer-colloid (copy archived at swh:1:rev:c3a89c-14cd928d37717d5f7a74cc1d1115b4bb7d). Figure 1—source code 1: Simulation parameters for the classical density functional theory code. Figure 3—source code 1: Simulation parameters for the classical density functional theory code.

The following dataset was generated:

| Author(s) | Year | Dataset title | Dataset URL | Database and Identifier |
|---|---|---|---|---|
| Davis LK | 2021 | DFT-polymer-colloid | https://github.com/patherlkd/DFT-polymer-colloid | Github, 164454451 |

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
