## [Editor Report]

This theoretical study describes the interaction of a planar brush or film of the resident unstructured components of the nuclear pore complex (NPC) called nucleoporins (FG-nups) and different nuclear transport receptors (NTRs). The authors describe impacts of competitive binding that give rise to enrichment of the NTRs, NTF2 and importin-β, at different depths of the FG-nup film, which could relate to experimental observations in other studies, as well as evidence that crowding could promote the rate of nuclear transport by modulating FG-NTR binding/unbinding. The conclusions were found to be generally supported by the data, relevant to the field of nuclear transport, and able to make specific predictions that can be experimentally tested in the future.

---

## [Decision Letter]

**Decision letter after peer review:**

Thank you for submitting your article "Crowding-induced phase separation of nuclear transport receptors in FG nucleoporin assemblies" for consideration by *eLife*. Your article has been reviewed by 3 peer reviewers, and the evaluation has been overseen by a Reviewing Editor and Vivek Malhotra as the Senior Editor. The reviewers have opted to remain anonymous.

Essential revisions:

Related to the Introduction and context:

1. The experimental system should be introduced: NTF2, importin β and Nsp1.

2. The authors should state the motivation for the study more clearly. Although they updated their model from single- to two-type NTR descriptions, their actual motivation was somewhat unclear. By knowing the binding affinity of one NTR in the presence of another NTR, what are they trying to prove? And what is the implication of this to understand the existing system of the NPC? Are there any phenomena that directly relate the findings of this study to the molecular transport at the NPC? They should elaborate the introduction and the Discussion section with these.

3. The manuscript should describe (with appropriate citations) how specific competitors and inert molecules affect relevant interactions within the NPC (e.g., Tetenbaum-Novatt, Mol Cell Proteomics, 2012 and many others). There are also some preceding works related to this study investigating the free energy landscape associated with the NTR-FG-Nups bindings that should be discussed including recent studies carried out in an NPC-like geometry (Sirkin et al., Soft Matter, 2021; Matsuda et al., Biophys J, 2021) as well as a planar surface (Lennon et al., Int. Jour. Mol. Sci., 2021); the authors should introduce this work for context but also relate their findings to these works in the Discussion.

4. Relating the work to other studies of competitive binding and protein adsorption in polymer brushes should be expanded, including a prior study by Szleifer and Fang and related works (Fang F, Szleifer I. Competitive adsorption in model charged protein mixtures: equilibrium isotherms and kinetics behavior. The Journal of Chemical Physics. 2003 Jul 8;119(2):1053-65.). Figure 5 in that work shows that for the adsorption of proteins on an oppositely charged surface, small proteins adsorb near the surface and large ones adsorb on top of the small ones. As this is the same behavior observed by the authors, whether the authors feel that both observations share a common origin should be addressed also in the Discussion.

5. Throughout the manuscript the authors should include major assumptions and simplifications vs. the native NPC when discussing the model.

Related to the approach:

6. An important feature is the parametrization of the model. However, this is not novel because the parametrization of a very similar model was previously explored by the authors in Zahn et al., (2016). The values of the interaction parameters are different in both papers, which likely results from different definitions for the interaction parameters. In this context, please clarify the following sentence: "We note that the value of \epsilonFG-FG found here is different to that of our previous work Zahn et al., (2016), which is due to the different choices of interaction potential and geometry" given that the interaction potential can change the value of the interaction parameters, but the geometry of the brush should not be a relevant factor.

7. Please define each free energy term by the mathematical expression. While F_{ideal-gas} and F_{ideal-polymer} are defined using equations, F_{external}, F_{exchanges}, F_{cohesion}, and F_{hard-shere} do not have formal definitions and then equation (3) appears suddenly. It is better to bring equation (4)-(6) before equation (3) to define each term of free energy and give more detailed explanations.

8. In line 175, they say, "Following previous work Zahn et al., (2016); Vovk et al., (2016), we do not consider any cohesion between NTRs themselves". The authors should explicitly explain why they did not include mutual interactions between NTRs since this model choice is critical for the result. For example, PMF in Figure 2 may not change so much along with NTR's concentration change if they included the mutual cohesion. More NTRs in the film would bring about the intermolecular attraction as well as the crowding effect.

9. A salient feature of the model is the use of a hard-sphere functional for the steric interactions. Two questions/comments related to this should be addressed: (1) The authors use hard-sphere (white bear) and chain connectivity functionals in their theory. A brief description of the physics behind each one (i.e., which interactions they model and how) should really help the reader to understand the approximations involved in the theory; and (2) The solvent is not explicitly included in the calculations. What would be the effect of adding a fourth hard-sphere fluid with a radius much smaller than that of the proteins to model solvent molecules?

Related to the interpretations:

10. According to studies by Lim, Peters and many others, NTRs are essential part of the selectivity barrier. (The authors have even cited Lim's 2015 work in the discussion). Does the potential barrier for inert molecules increase in the presence of NTRs?

11. Several important points were raised regarding how the intersection of NTR size, NTR-FG interactions, and NTR-cargo interactions impact the results. According to the authors, "layering effects" depend on the size of NTRs. Given that the number of binding pockets on the two modeled NTRs is significantly different (9 vs. 2), it is unclear whether the effects of avidity have been fully considered. Could avidity effects play an additional role in layering? Further, there is no examination of how the cargo bound to the NTR will impact the described behavior. For NTF2 the sole cargo (Ran-GDP) could be considered in a straightforward manner compared to the diverse cargos for importin β; nonetheless the small size of Ran would seem to perhaps exacerbate the trend suggested by the modeling studies?

12. On a related note, it was felt that there needed to be greater exploration of the phase separation of NTRs. For example, what is the main factor that triggered the phase separation? Is it the difference in the NTR size? Or is it the difference in the cohesion energy? In the manuscript, the authors consider the existence of the small NTRs as the important factor determining the phase separation. Then, how small should the NTR be to create this phenomenon? The authors can either run more simulations to explore the critical diameter of the NTR creating the phase separation or at least add some comment about why the size of NTF2 was good enough for the phase separation.

13. There was a consensus that performing additional calculations for a system with two opposing surfaces would strengthen the relevance of the work. It was expected that in a DFT approach that should not involve too much effort (it probably may be implemented by changing boundary conditions) and it should provide at least some insight on the effect of nanoconfinement. Calculations for a cylindrical geometry would be even better, but at least the easier task of two surfaces should be achievable.

Related to the Discussion:

14. The authors should clarify how their current studies advance their previous work (Zahn et al.,). The differences between the two should be clearly stated. For example, eFG-FG is parametrized in both manuscripts based on the same experimental data. While the authors note differences in both the selected interaction potential and geometry of the film assembly, it is not clear if those differences alone provide new results and conclusions. What drives the innovation in the approach of this manuscript? Overall, the manuscript could be enhanced if the authors stated more clearly both the assumptions and simplifications they made when they performed their analyses.

15. The model of NTR organization should be better compared to other proposed models (e.g., Wagner at al. Biophysics J. 2015.)

---

## [Author Response]

Essential revisions:Related to the Introduction and context:1. The experimental system should be introduced: NTF2, importin β and Nsp1.

We have now more explicitly introduced the system of study. We also note that this manuscript has been submitted as a *Research Advance* related to *Zahn et al.,* (*2016*). For this type of manuscript, the *eLife* guidance stipulates that “there may only need to be minimal introductory material”. Hence we feel that in response to this comment, it may be appropriate to also refer to *Zahn et al.,* (*2016*), where NTF2, Importin-*_* and Nsp1 are introduced in greater detail.

2. The authors should state the motivation for the study more clearly. Although they updated their model from single- to two-type NTR descriptions, their actual motivation was somewhat unclear. By knowing the binding affinity of one NTR in the presence of another NTR, what are they trying to prove? And what is the implication of this to understand the existing system of the NPC? Are there any phenomena that directly relate the findings of this study to the molecular transport at the NPC? They should elaborate the introduction and the Discussion section with these.

Briefly, we aim to understand how crowding with different types of NTRs affects binding of NTRs in FG Nup assemblies. This is relevant for molecular transport in the NPC because it provides a means to modulate specific transport by competitive binding of different NTRs and/or to create separate pathways through the NPC based on phase-separation between different types of NTRs.

We have now revised the introduction and discussion to make this clearer.

3. The manuscript should describe (with appropriate citations) how specific competitors and inert molecules affect relevant interactions within the NPC (e.g., Tetenbaum-Novatt, Mol Cell Proteomics, 2012 and many others). There are also some preceding works related to this study investigating the free energy landscape associated with the NTR-FG-Nups bindings that should be discussed including recent studies carried out in an NPC-like geometry (Sirkin et al., Soft Matter, 2021; Matsuda et al., Biophys J, 2021) as well as a planar surface (Lennon et al., Int. Jour. Mol. Sci., 2021); the authors should introduce this work for context but also relate their findings to these works in the Discussion.

We have revised the introduction and Discussion sections to include such a description and appropriate citations.

4. Relating the work to other studies of competitive binding and protein adsorption in polymer brushes should be expanded, including a prior study by Szleifer and Fang and related works (Fang F, Szleifer I. Competitive adsorption in model charged protein mixtures: equilibrium isotherms and kinetics behavior. The Journal of Chemical Physics. 2003 Jul 8;119(2):1053-65.). Figure 5 in that work shows that for the adsorption of proteins on an oppositely charged surface, small proteins adsorb near the surface and large ones adsorb on top of the small ones. As this is the same behavior observed by the authors, whether the authors feel that both observations share a common origin should be addressed also in the Discussion.

We thank the reviewers for referring us to this very relevant work, which indeed can be explained in a similar way. We have revised our manuscript and particularly the Discussion section accordingly.

5. Throughout the manuscript the authors should include major assumptions and simplifications vs. the native NPC when discussing the model.

We have revised the manuscript accordingly.

Related to the approach:6. An important feature is the parametrization of the model. However, this is not novel because the parametrization of a very similar model was previously explored by the authors in Zahn et al., (2016). The values of the interaction parameters are different in both papers, which likely results from different definitions for the interaction parameters. In this context, please clarify the following sentence: “We note that the value of \epsilonFG-FG found here is different to that of our previous work Zahn et al., (2016), which is due to the different choices of interaction potential and geometry" given that the interaction potential can change the value of the interaction parameters, but the geometry of the brush should not be a relevant factor.

This comment is correct and we have revised this sentence accordingly. The main difference is due to the different definitions for the interaction parameters, with some minor – and indeed not relevant – differences due to the boundary conditions imposed in *Zahn et al.,* (*2016*).

7. Please define each free energy term by the mathematical expression. While F_{ideal-gas} and F_{ideal-polymer} are defined using equations, F_{external}, F_{exchanges}, F_{cohesion}, and F_{hard-shere} do not have formal definitions and then equation (3) appears suddenly. It is better to bring equation (4)-(6) before equation (3) to define each term of free energy and give more detailed explanations.

We have revised the computational model section accordingly.

8. In line 175, they say, "Following previous work Zahn et al., (2016); Vovk et al., (2016), we do not consider any cohesion between NTRs themselves". The authors should explicitly explain why they did not include mutual interactions between NTRs since this model choice is critical for the result. For example, PMF in Figure 2 may not change so much along with NTR's concentration change if they included the mutual cohesion. More NTRs in the film would bring about the intermolecular attraction as well as the crowding effect.

We have assumed that the NTRs do not experience direct attractive interactions between them, since we are not aware of experimental evidence to suggest otherwise. We have now made that more explicit in the manuscript. The PMFs in Figure 2 are directly related to the experimentally validated total NTR concentrations in the film *Zahn et al.,* (*2016*), which are well described by our minimal model in the absence of attractive interactions between NTRs.

9. A salient feature of the model is the use of a hard-sphere functional for the steric interactions. Two questions/comments related to this should be addressed: (1) The authors use hard-sphere (white bear) and chain connectivity functionals in their theory. A brief description of the physics behind each one (i.e., which interactions they model and how) should really help the reader to understand the approximations involved in the theory; and (2) The solvent is not explicitly included in the calculations. What would be the effect of adding a fourth hard-sphere fluid with a radius much smaller than that of the proteins to model solvent molecules?

1) Now in the methods: Essentially, the White bear functional removes – from the free energy – contributions from system configurations with particle overlaps, which is the definition of a hard-sphere system. The hard-sphere functional is built using a standard formalism known as fundamental measure theory, attributed to Rosenfeld, which begins with a decomposition of the Mayer function into weight functions that contain geometrical information of spherical particles (see *Rosenfeld* (*1989*); *Roth et al.,* (*2002*); *Roth* (*2010*) for details). The chain-connectivity function is built in a similar manner, but removes from the free energy contributions from system configurations that violate how consecutive beads on a chain should be connected, as specified by the particular polymer model (see *Yu and Wu* (*2002*) for more details).

2) Indeed, the solvent is implicit as now mentioned in the methods. Here, we do not explicitly describe the small-scale solvent molecules as the dominant interactions come from the NTRs and the FG Nups. However the solvent medium is implicitly accounted for when specifying the intra and inter-actions between FG Nups and the interactions between NTRs and FG Nups. This can be illustrated by the packing fractions of FG Nups and NTRs as derived from experiments *Zahn et al.,* (*2016*), occupying *<* 20~ of the film volume: The difference in solvent contents between bulk solution (close to 100%) and film (*>* 80~) is relatively small.

Related to the interpretations:10. According to studies by Lim, Peters and many others, NTRs are essential part of the selectivity barrier. (The authors have even cited Lim's 2015 work in the discussion). Does the potential barrier for inert molecules increase in the presence of NTRs?

To answer this question, we have carried out additional simulations and included the new Figure 2—figure supplement 1, and revised the text of our manuscript accordingly. For the largest inert particles considered here (diameter *d*
_= 6_ nm), we find an increase of the potential barrier of several *k*_B_*T* due to NTF2 binding in a physiologically relevant concentration range, in agreement with the notion that NTRs contribute to the integrity of the NPC transport barrier.

11. Several important points were raised regarding how the intersection of NTR size, NTR-FG interactions, and NTR-cargo interactions impact the results. According to the authors, "layering effects" depend on the size of NTRs. Given that the number of binding pockets on the two modeled NTRs is significantly different (9 vs. 2), it is unclear whether the effects of avidity have been fully considered. Could avidity effects play an additional role in layering? Further, there is no examination of how the cargo bound to the NTR will impact the described behavior. For NTF2 the sole cargo (Ran-GDP) could be considered in a straightforward manner compared to the diverse cargos for importin β; nonetheless the small size of Ran would seem to perhaps exacerbate the trend suggested by the modeling studies?

Could avidity effects play an additional role in layering? Is a very interesting question that, in short, we don’t know the exact answer to and that – without further experimental data – would be rather non-trivial to parametrize in our minimal, homogeneous-spherical NTR model. Similarly, we can only consider the effect of, e.g., Ran-GDP binding to NTF2 (resulting in a complex with a larger size than NTF2 alone) by exploring how the observed phase separation depends on relative sizes of the NTRs. We have carried out additional simulations for this purpose (the new Figure 5 and associated Figure supplements) and amended our Discussion section to consider the effects of avidity and bound cargo as far as possible within the context of our minimal model.

12. On a related note, it was felt that there needed to be greater exploration of the phase separation of NTRs. For example, what is the main factor that triggered the phase separation? Is it the difference in the NTR size? Or is it the difference in the cohesion energy? In the manuscript, the authors consider the existence of the small NTRs as the important factor determining the phase separation. Then, how small should the NTR be to create this phenomenon? The authors can either run more simulations to explore the critical diameter of the NTR creating the phase separation or at least add some comment about why the size of NTF2 was good enough for the phase separation.

This is an excellent suggestion that we have followed up via the new simulations now represented in the new Figure 3—figure supplement 2, Figure 5—figure supplements 1 and 2, summarised in the new Figure 5, and discussed in the revised text.

13. There was a consensus that performing additional calculations for a system with two opposing surfaces would strengthen the relevance of the work. It was expected that in a DFT approach that should not involve too much effort (it probably may be implemented by changing boundary conditions) and it should provide at least some insight on the effect of nanoconfinement. Calculations for a cylindrical geometry would be even better, but at least the easier task of two surfaces should be achievable.

We agree with the reviewers that moving to a geometry that more closely resembles the NPC would be a natural next step, to more precisely probe the effects of nanoconfinement in a pore system; and that a system with two opposing surfaces could add relevant information, if properly calibrated. Unfortunately though, such a proceeding is not as straightforward as suggested by this comment. Firstly, at present the boundary conditions are hard-coded into our DFT algorithms, such that its extension to two opposing surfaces represents substantial effort in terms of technical development and validation. Secondly, it is not trivial to determine which inter-surface distance would yield behaviour that could be compared with the NPC pore, all the more since qualitatively different types of behaviour can occur even for FG Nups alone in a pore geometry *Hoogenboom et al.,* (*2021*) depending on the parameter settings.

To address this comment, we have instead revised our manuscript at several places to most explicitly articulate the limitations and caveats of our approach and of its extrapolation to the real NPC, while also emphasising the strengths of the planar FG Nup assemblies as a model system to study NPC functionality. In particular, these FG Nup films reproduce – in experiments and in computational simulations – the binding, penetration and/or relative exclusion of NTRs and inert particles in a way that is fully consistent with the functional behaviour observed for NPCs in vivo.

Related to the Discussion:14. The authors should clarify how their current studies advance their previous work (Zahn et al.,). The differences between the two should be clearly stated. For example, eFG-FG is parametrized in both manuscripts based on the same experimental data. While the authors note differences in both the selected interaction potential and geometry of the film assembly, it is not clear if those differences alone provide new results and conclusions. What drives the innovation in the approach of this manuscript? Overall, the manuscript could be enhanced if the authors stated more clearly both the assumptions and simplifications they made when they performed their analyses.

We have now amended the (beginning of the) discussion to clarify this better.

15. The model of NTR organization should be better compared to other proposed models (e.g., Wagner at al. Biophysics J. 2015.)

We have revised the discussion to include such an improved comparison.